# Vermiform Appendix within Post-Laparoscopic Incisional Hernia: A Unique Case Report and Literature Review

**DOI:** 10.3390/medicina59030538

**Published:** 2023-03-10

**Authors:** Kristina Marcinkeviciute, Gabija Makunaite, Donatas Danys, Kestutis Strupas

**Affiliations:** 1Faculty of Medicine, Vilnius University, LT-03101 Vilnius, Lithuania; 2Center of Abdominal Surgery, Clinic of Gastroenterology, Nephrourology, and Surgery, Institute of Clinical Medicine, Faculty of Medicine, Vilnius University, LT-03101 Vilnius, Lithuania

**Keywords:** vermiform appendix, incisional hernia, port-site incisional hernia, laparoscopic hernia repair

## Abstract

*Background*: Appendicitis within incisional hernia is an extraordinarily rare postoperative complication with an incidence range from 0.08 to 1%. From the 14 cases that we found in the English literature, only three present appendixes vermiform in incisional hernia followed by laparoscopic surgery. Only two cases are treated minimally invasively by the laparoscopic approach. *Case presentation*: We introduce a 65-year-old man who had a laparoscopic sigmoid colon resection and had a lump found at the 12 mm trocar site in the right iliac area in the late postoperative phase. There were no complaints from the patient. A vermiform appendix was unexpectedly discovered in the sac of that incisional hernia during control CT scans performed by chemotherapists. Laparoscopic hernia repair without appendectomy was performed. Postoperative outcomes were excellent. *Conclusions*: Because of low incidence and a lack of distinctive clinical presentation of appendicitis within incisional hernia, there is a risk of delayed perioperative diagnosis and treatment. A CT scan might play an important role in verifying the diagnosis early. For better postoperative outcomes, if possible, laparoscopic surgery should be chosen.

## 1. Introduction

An incisional hernia is the most frequent surgical complication, with an estimated frequency of 35% [1]. However, the incidence of an inflamed appendix within a hernial sac is very low and ranges from 0.08 to 1% [2]. From these, appendicitis within an inguinal or inguinoscrotal hernia is the most prevalent [3]. Only very few papers have been published reporting appendicitis within incisional hernias. We counted only 14 such cases worldwide in the literature (Table 1) and only 3 of them were in incisions of previous laparoscopic operations [4,5,6]. In fact, in all of these three mentioned cases, the vermiform appendix was inflamed and none of them followed the laparoscopic approach of hernioplasty [6]. According to this statistic, we present an extraordinary case of a non-inflamed appendix within an incisional hernia after laparoscopic sigmoid colon resection, which was repaired in a minimally invasive way.

## 2. Case Report

The patient gave their consent for this piece of research. Approval from the Institutional Review Board was granted.

We report a 65-year-old male patient with an appendix in an unusual post-laparoscopic hernia in the 12 mm trocar site in the right iliac fossa. He had only surgical history of laparoscopic sigmoid colon resection for sigmoid colon carcinoma (pT1N0M0) 3 years prior. In the early postoperative course, it went without complications. After that, the patient continued to be treated and monitored by chemotherapists. In the late postoperative period, a lump was detected at the 12 mm trocar site in the right iliac region of a previous surgery site that was observed as an incisional hernia. However, the patient had no complaints. During control CT scans by chemotherapists, a vermiform appendix was accidentally observed in the sac of that incisional hernia (Figure 1). Laboratory tests were normal, and the patient still had no other symptoms, except the lump. There were no signs of inflammation—clinically or radiologically; it was decided to perform a planned operation by the laparoscopic approach. Three trocars were placed in the abdominal cavity.

Despite the absence of symptoms, it was decided to operate due to the contents in the hernia sac. During the operation, it was observed that the size of the hernial defect was 12 mm. The vermiform appendix and the mesentery had entered the incision of the trocar in the right iliac region (Figure 2). They were easily returned to the abdominal cavity; the adhesion fixing the appendix was cut and the aponeurosis here was sutured using a Berci needle with two stitches of Prolene. There were no indications for appendectomy and it was not performed. A hernia mesh was not used due to the small size of the defect, as well as due to the absence of risk factors such as obesity or heavy physical work. After this operation, the patient’s condition was excellent, he had no complaints, and inflammatory indicators were within normal limits, so the next day he was discharged from the hospital. Early cosmetic results were also excellent (Figure 3). After a 5-week follow-up, the patient had no complaints and there were no signs of early relapse.

## 3. Discussion

The incidence of port site hernia following laparoscopic surgery ranges from 0.38% to 5.4%, with a total incidence of 1.7% [17]. The bigger risk for hernia development is 10 mm and larger in diameter trocar incisions [18]. Although the location of the appendix in the abdominal cavity can be quite variable, its detection in a hernial sac is extremely rare [7]. In our case, the hernia with the appendix appeared at the location of a 12 mm trocar, which we routinely use for laparoscopic sigmoid colon resections. There is one more similar case in the literature, where in the same location incision, a hernia with an appendix inside of it formed after low rectum resection [6]. In another case report, a hernia with appendicitis in the hernia sac developed at the site of an umbilical 10 mm incision [4]. Even though the probability of postoperative incisional hernia formation at a site where an incision of less than 10 mm was made is extremely low, there is one case described in the literature where this phenomenon was observed at the site of a 5 mm trocar incision [5].

Appendicitis typically begins with widespread or periumbilical abdominal discomfort that radiates to the right lower quadrant and is usually followed by anorexia, nausea, vomiting, and fever [19]. Abdominal ultrasound evaluation due to its excellent specificity and sensitivity for diagnosing acute appendicitis is now generally accepted as a standard [14]. In situations with incisional hernia appendicitis, the classic appearance of appendicitis may be missing [11]. That leads to a worse prognosis of this situation because the diagnosis of appendicitis may be delayed and it might progress to a gangrenous form and even perforate, causing a life-threatening condition [8]. In these complicated cases, when appendicitis is located in the sac of the hernia, the diagnosis is usually verified by CT by suspecting other pathologies because of the unusual clinical presentation of appendicitis [10]. The clinical presentation of the reported examples in the literature is similar to that of an incarcerated hernia [11]. After a complicated hernia is identified, it is not standard procedure to submit the patient to a CT scan. However, our case shows that while the appendix is not inflamed, the symptoms might not appear and it can only be detected accidentally.

The preferred treatments for appendicitis within incisional hernia are appendectomy and hernia repair utilizing laparoscopic or open techniques [10]. The treatment of a non-inflamed appendix in the hernia sac is still controversial [9]. Most studies suggest surgical treatment tactics even though there are not enough clinical or radiological signs of acute appendicitis, while some others prefer avoiding operations due to possible complications if there is no data for inflammation of the appendix [11,13,16]. Since such cases are very few, there is a limitation of proof of the superiority of either choice. We did not perform an appendectomy because of the absence of signs of acute appendicitis: the laboratory tests and radiologic views seemed normal and when examined during laparoscopy, they did not appear altered. It was decided to avoid the possible complications after appendectomy—an intra-abdominal abscess, bleeding from the caecum, or appendix burst during the operation [20,21]. Moreover, the appendix has been established as an important component not only for the passage and elimination of waste matter in the lower digestive tract but also as part of the organism of stimulating immunity due to B- and T-lymphocyte-mediated response [22]. After performing any intervention in the abdominal cavity, where the anatomy is damaged, including appendectomy, adhesions form as a result [23]. Therefore, any other intervention would be more complicated due to adhesions.

During the same operation, not only appendix removal should be considered but also hernioplasty to fix the hernia. For the repair of a hernia defect, it is generally recommended to use a hernia mesh to restore the integrity of the abdominal wall through open or laparoscopic surgery. Over the years, the discussion in the literature about which method is superior is still continuing. In recent years, there has been an increase in the advantages of laparoscopic surgery in the postoperative period after hernia surgery [24]. In our case, surgery was not technically difficult to perform when the appendix was not inflamed and there were no indications to remove it. We chose laparoscopic surgery for hernia repair with all the advantages of minimally invasive treatment, less pain, shorter hospitalization time, faster healing, and better cosmetic outcomes. There are some surgical choices for abdominal wall defect closure: an intraperitoneal onlay mesh (IPOM), IPOM-plus (when the fascia defect is sutured before the mesh insertion), transabdominal preperitoneal (TAPP) repair, or intracorporeal suturing of the defect [25,26,27]. There is no ideal method; therefore, the knowledge of a wide range of surgical choices applied by surgeons of various skill levels is the ideal solution. The size of the hernia defect mainly determines the surgeon’s choice. The laparoscopic approach for incisional hernias bigger than 15 cm is challenging, with a significant complication rate as a result for these patients’ open hernia repair techniques (the onlay, sublay (retromuscular), or inlay methods) are recommended [28,29]. For smaller defects’ repair, laparoscopic techniques could be considered. We did not use the mesh for hernia repair because of a small defect (only 12 mm) for which two stitches of Prolene were enough for its closure. Furthermore, when the hernia gap is smaller than 2 cm, the study [27] showed that hernia defect suturing without using a mesh is safe and the recurrence is low (4.1%). However, if in our case the defect size were bigger, the mentioned alternatives should have been considered.

To prevent incisional hernias, trocar sites with fascial defects of 10 mm or more, including the peritoneum, should be closed [30]. Opinions differ in the literature on whether a 5 mm trocar site defect should be closed [30,31]. Some studies report that in cases of paramedian locations and blunt-type trocars (conical, pyramidal, radially dilating, and non-bladed) usage, fascial closure is not crucial not only for 5 mm but also for 10 mm and 12 mm incisions [31]. However, most of the studies do not provide recommendations for 5 mm trocar site defect fascial closure [31]. Blunt-type trocars used for 5 mm trocar sites split the muscles instead of cutting them and in this way reduce the area of the facial defect [32,33]. The defect of 5 mm trocar insertion should be closed only if the defect might be extended due to manipulation in the abdomen [31].

## 4. Conclusions

Appendicitis in an incisional hernia is an extremely rare occurrence. Because of the low frequency and unique clinical presentation, the risk of possible consequences owing to delayed perioperative diagnosis and treatment is significant. All clinicians should evaluate it to avoid the consequences of delayed surgical treatment. CT scans might aid in early diagnosis. In addition, minimally invasive surgery should be considered. To avoid appendix vermiform or other structures within the incisional hernia after laparoscopic surgery, 10 mm and bigger abdominal defects should be closed.

## Figures and Tables

**Figure 1 medicina-59-00538-f001:**
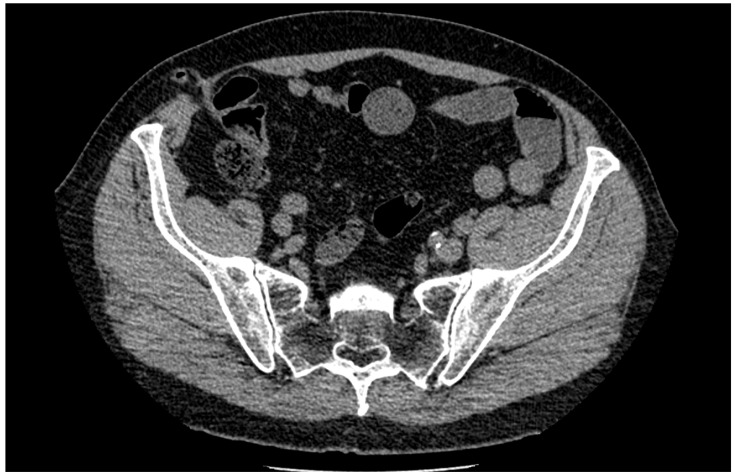
A vermiform appendix in the sac of the incisional hernia on a computed tomography scan.

**Figure 2 medicina-59-00538-f002:**
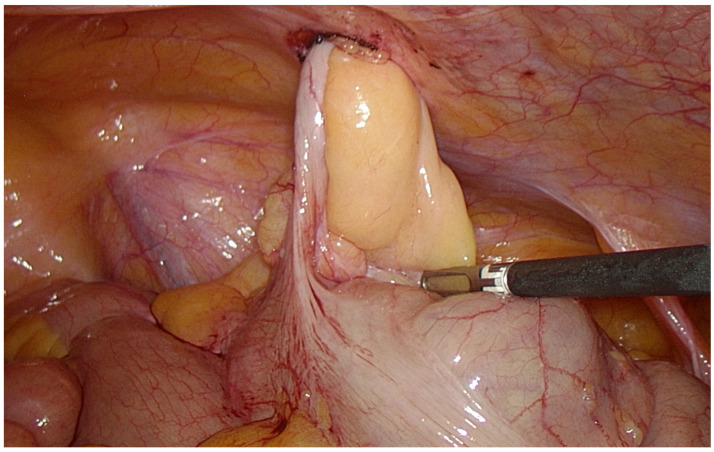
Vermiform appendix together with the mesentery within the previous trocar incision in the right iliac region.

**Figure 3 medicina-59-00538-f003:**
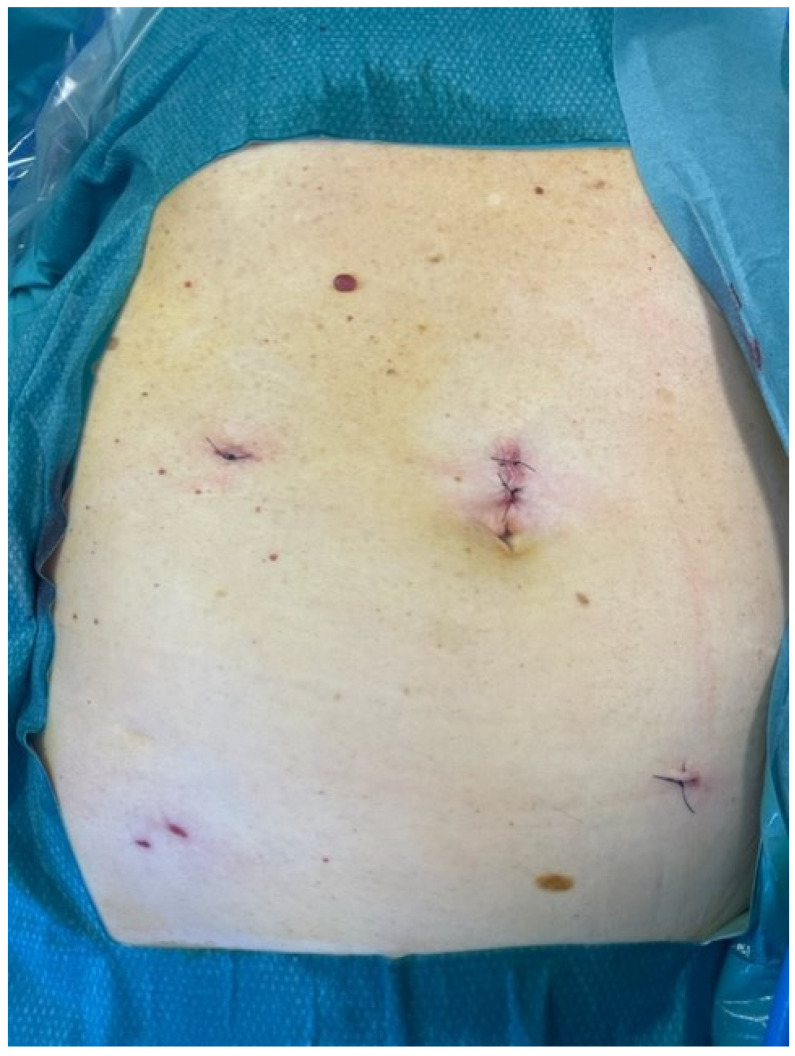
Early cosmetic results after the operation.

**Table 1 medicina-59-00538-t001:** Previous cases of appendix within the incisional hernia.

No	Authors	Previous Operation	Location of the Hernia	Type of Performed Operation	Normal or Altered Appendix
1	Erol T. et al. [7]	NA	NA	NA	Inflamed
2	Paudyal N. et al. [8]	Laparotomy with right-sided salpingectomy	Pfannenstiel incision	Open surgery	Inflamed
3	Lakhani DA. et al. [9]	Left total nephrectomy and renal transplantation	Right lower quadrant incisional hernia	Open surgery	Perforated
4	Molina G. et al. [10]	Open cholecystectomy	Kocher incision	Open surgery	Inflamed
5	Lam A. et al. [4]	Laparoscopic sterilization	Umbilical laparoscopic port site incision	Open surgery	Inflamed
6	Kler A. et al. [11]	Open total hysterectomy	Pfannenstiel incision	Open surgery	Normal
7	West C. et al. [12]	Open abdominal aortic aneurysm repair	Laparotomy incision	Open surgery	NA
8	Sugrue C. et al. [5]	Open cholecystectomy	Upper midline incisional hernia	Open surgery	Inflamed
9	Sugrue C. et al. [5]	Diagnostic laparoscopy and lavage	Five mm port site in the right iliac fossa	Open surgery	Inflamed
10	Galiñanes EL. et al. [13]	Total abdominal hysterectomy with right-sided oophorectomy	Pfannenstiel incision	Laparoscopic surgery	Inflamed
11	Dittmar Y. et al. [14]	Kidney transplantation	Right lower quadrant incisional hernia	Laparoscopic surgery	Inflamed
12	Singal R. et al. [15]	Open surgery of subsequent bone grafting from the right iliac crest	Previous operation incision	Open surgery	Inflamed
13	Menenakos Ch. et al. [6]	Laparoscopic low anterior rectal resection	Twelve mm trocar site incision in the right iliac fossa	Open surgery	Inflamed
14	McKay DW. et al. [16]	Open cholecystectomy	Kocher incision	Open surgery	Normal

NA—not available.

## Data Availability

Data is contained within the article.

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
