# Peer review of "Vermiform Appendix within Post-Laparoscopic Incisional Hernia: A Unique Case Report and Literature Review"

_medicina, 2023, doi:10.3390/medicina59030538_

Round 1

Reviewer 1 Report

This is a very curious case successfully managed by a minimal invasive surgical approach. 

Very few case reports being published, this is a good manuscript but it clearly needs extensive English editing.  

Author Response

Dear reviewer,

Thank you for your comment. We understood your concern that there was a need to edit the text. We have revised it and made corrections (please see the track changes in the attachment).

Reviewer 2 Report

Thank you for giving this ooprtunity to me to review this interesting case report. In my opinion, this is a very interesting and rare case report.

 The authors conducted a case review of the literature. The figures are quite good image that shows the case. It can be published.

Author Response

Dear Reviewer,

Thank you for your comments. 

Reviewer 3 Report

The authors presented a rare case in an appropriate way.

Apart from the way in which the abdominal wall defect was solved, there are other operative techniques that should be mentioned, such as IPOM and IPOM plus technique, then TAP, also intracorporeal suturing of the defect, and so on.

My question for the authors is why appendectomy was not performed according to the findings - a few words should be written about it. Why would the patient expose himself to a potentially new operation in the future, when the results were already like this?

A few words are missing with literature data on whether the defect should be reconstructed after the placement of a 10mm or 12mm port. Yes or No - views from current literature

Author Response

Dear editor,

Thank you for your comment. We understood your concern that there were not mentioned other surgical techniques for hernia repair (such as IPOM and IPOM plus technique, then TAP, also intracorporeal suturing of the defect), so we have discussed them now in our text.

We did not perform an appendectomy because of the absence of signs of acute appendicitis: the laboratory tests, radiologic views seemed normal and when examined during laparoscopy, it did not appear altered. It was decided to avoid the possible complications after the appendectomy – intra-abdominal abscess, bleeding from the caecum, or appendix burst during the operation. Moreover, the appendix has been established to have some functions: it is useful for the passage and elimination of waste matter in the lower digestive tract and according to the literature (references in the text) is a part of the system of stimulating immunity due to lymphocytes B and T mediated response. After performing any intervention in the abdominal cavity, where the anatomy is damaged, including appendectomy, the adhesions form as a result. Therefore, any other intervention would be more complicated due to adhesions. For all these reasons operating surgeons decided not to perform appendectomy along with hernia repair operation.

We have checked the literature on fascial defect closure of 5 mm, 10 mm, or 12mm ports. According to the data in the literature, all 10 mm and bigger (12 mm) ports should be closed. Some studies report the exception for 10 mm and 12 mm ports in cases of paramedian locations and blunt-type trocars (conical, pyramidal, radially dilating, non-bladed) usage.

See the details in the attachment,.
